# miR-210 Expression Is Strongly Hypoxia-Induced in Anaplastic Thyroid Cancer Cell Lines and Is Associated with Extracellular Vesicles and Argonaute-2

**DOI:** 10.3390/ijms24054507

**Published:** 2023-02-24

**Authors:** Bonita H. Powell, Andrey Turchinovich, Yongchun Wang, Olesia Gololobova, Dominik Buschmann, Martha A. Zeiger, Christopher B. Umbricht, Kenneth W. Witwer

**Affiliations:** 1Department of Molecular and Comparative Pathobiology, Johns Hopkins University School of Medicine, Baltimore, MD 21205, USA; 2Division of Cancer Genome Research, German Cancer Research Center (DKFZ), 69120 Heidelberg, Germany; 3Heidelberg Biolabs GmbH, 69120 Heidelberg, Germany; 4Department of Surgery, Johns Hopkins University School of Medicine, Baltimore, MD 21205, USA; 5Center for Cancer Research, National Cancer Institute, National Institute of Health, Bethesda, MD 20892, USA; 6Department of Neurology, Johns Hopkins University School of Medicine, Baltimore, MD 21205, USA

**Keywords:** microRNA, miRNA, miR-210, hypoxia, HIF1-alpha, extracellular vesicles, anaplastic thyroid cancer

## Abstract

Hypoxia, or low oxygen tension, is frequently found in highly proliferative solid tumors such as anaplastic thyroid carcinoma (ATC) and is believed to promote resistance to chemotherapy and radiation. Identifying hypoxic cells for targeted therapy may thus be an effective approach to treating aggressive cancers. Here, we explore the potential of the well-known hypoxia-responsive microRNA (miRNA) miR-210-3p as a cellular and extracellular biological marker of hypoxia. We compare miRNA expression across several ATC and papillary thyroid cancer (PTC) cell lines. In the ATC cell line SW1736, miR-210-3p expression levels indicate hypoxia during exposure to low oxygen conditions (2% O_2_). Furthermore, when released by SW1736 cells into the extracellular space, miR-210-3p is associated with RNA carriers such as extracellular vesicles (EVs) and Argonaute-2 (AGO2), making it a potential extracellular marker for hypoxia.

## 1. Introduction

Anaplastic thyroid carcinoma (ATC) is a rare malignancy, yet it accounts for the majority of all thyroid tumor-related deaths [1,2,3,4]. In contrast with thyroid cancers such as papillary thyroid cancer (PTC), ATC cases are characteristically aggressive and almost always fatal, as these tumors acquire numerous deleterious genetic aberrations during de-differentiation [1,5]. Anaplastic thyroid carcinomas, as with many other types of solid tumors, are highly proliferative and often hypoxic (oxygen-deprived) [5,6,7,8]. Tumors adapt to hypoxic stress by activating various cell signaling pathways through the action of hypoxia-inducible factors (HIFs). HIF-1 is a heterodimer composed of an alpha and a beta subunit [9,10]. In normoxic conditions, HIF-1α is produced constitutively but is rapidly degraded by the proteasome upon hydroxylation. This occurs at specific proline residues by prolyl hydroxylase (PHD), followed by ubiquitination by the von Hippel–Lindau (VHL) protein [11,12,13]. In hypoxic conditions, HIF-1α proteins remain stable and translocate to the nucleus to activate genes by associating with the HIF-1ß subunit and binding to HIF Response Elements (HREs) within the proximal promoters of target genes [14,15,16]. Several HIF targets are involved in cell cycle regulation, metabolism and angiogenesis [14,17,18,19,20,21,22]. Furthermore, hypoxic tumor cells with elevated HIF-1α protein levels are often resistant to chemotherapy and ionizing radiation [8,23,24]. Hence, developing approaches to identify and target hypoxic cells within ATC tumors may improve patient outcomes.

miRNAs are short, ~22 nucleotide, post-transcriptional gene regulators that bind to complementary regions within the 3′UTR of mRNAs, thereby inhibiting translation or mediating degradation [25,26,27,28]. Cells release miRNAs into the extracellular milieu, where they are protected from RNase digestion in carriers including extracellular vesicles (EVs) and extracellular proteins such as Argonaute-2 (AGO2) [29,30,31,32,33,34,35]. Thus, miRNAs may be particularly attractive for the purpose of “liquid biopsies”: using biological fluids to assess the presence and state of cancer cells in the body. We and several others have reported dysregulated microRNAs (miRNAs) in thyroid cancers. The most commonly aberrantly expressed miRNAs in ATC include miR-222, miR-221, miR-146b and miR-21 [36,37,38,39,40,41]. However, there is a paucity of information on miRNA expression differences in hypoxic vs. non-hypoxic regions within thyroid tumors [30,31,32,33,34,35].

In this study, we examined miR-210-3p (“miR-210”) as a potential marker of hypoxia. miR-210 is a well-known hypoxia-responsive miRNA and a direct HIF-1 target [42,43,44,45]. Its expression is frequently up-regulated in hypoxic tumor cells. As with HIF-1α protein levels, elevated miR-210 expression is associated with a poor prognosis [46,47]. We report herein on miR-210 expression in both PTC and ATC cell lines, examine the influence of hypoxia on cellular and extracellular miR-210-3p in the ATC line SW1736 and assess the extracellular association of miR-210 with EVs and AGO2.

## 2. Results

### 2.1. miR-210-3p Expression Induced by HIF-1α Stabilization in SW1736

To confirm the inducible expression of miR-210-3p in response to HIF-1α stabilization in hypoxia, SW1736 (ATC) cells were cultured at 21% O_2_ (normoxia) or 2% O_2_ (hypoxia) for 24, 48, 72, 96 and 120 h. Increased HIF-1α protein levels in hypoxia were observed at each time point; the greatest increases compared with normoxic controls were at 24 and 48 h (Figure 1B). Concurrent with increased HIF-1α protein levels, miR-210-3p expression increased at each hypoxic time point, with the highest expression levels also at 24 h (~9-fold) and 48 h (~16-fold) relative to normoxia (Figure 1C). Both HIF-1α and miR-210-3p levels slightly declined by 72 h and thereafter (Figure 1B,C). This is consistent with our observation and previous reports of decreased levels of HIF-1α protein due to mRNA instability in prolonged hypoxia (Figure 1D) [48,49,50] as well as the negative regulation of HIF-1α by miR-210 in a study of human T-cells [51].

Additionally, cell morphology, viability and glucose consumption were compared in hypoxia vs. normoxia. Cells from each condition were visibly comparable (Figure 1E,F) and shared a similar viability, but cell counts were lower in hypoxia (Figure 1G). Additionally, glucose consumption significantly increased in hypoxia (Figure 1H). Taken together, our results confirm the positive correlation between miR-210-3p expression and HIF1-α protein levels in SW1736.

### 2.2. Hypoxia Induces miR-210 More Strongly in ATC Compared with PTC Lines

Small RNA sequencing was performed on ATC cell lines SW1736 and C643 and PTC cell lines BCPAP and TPC-1, all grown under normoxic culture conditions. We identified 28 miRNAs that were differentially expressed (fold-change > 2, *p*-value < 0.05) between normoxic ATC and PTC lines (Figure 2A). Under normoxia, miR-210 was ~5-fold less abundant in ATC lines versus PTC lines (Figure 2A). qPCR validation confirmed this result for normoxia (Figure 2B). However, in hypoxia, ATC miR-210 expression reached levels comparable to or greater than those in PTC cells (Figure 2B). A time course assay examining points from 0 to 48 h of hypoxia revealed that ATC cell lines showed a higher fold change, >2-fold, of miR-210-3p expression compared with the PTC cell lines (Figure 2C). Therefore, although basal (normoxic) levels of miR-210 are lower in ATC than PTC lines, the miRNA is more strongly up-regulated by hypoxia in ATC lines.

### 2.3. miR-210-3p Is the Primary Hypoxia-Responsive miRNA in SW1736 Cells

To examine how hypoxia might impact the expression of additional miRNAs in SW1736, small RNA-sequencing was performed to assess miRNA differences in normoxia vs. hypoxia at 72 h. Although miR-210-3p was up-regulated in hypoxia (~6.5-fold, *p* < 0.05), no other miRNAs were significantly differentially expressed greater than 2-fold, *p* < 0.05, except for the precursor premiRNA-210 (~2.5-fold, *p* < 0.05) (Figure 3A). Interestingly, most of the reads aligned to 3-*p* termini of pre-miR-210 were 1-2 nucleotides longer than the canonical mature miR-210 form and frequently contained mismatched bases at +1 location (T → G, C, A) and +2 location (C → T, G, A) (Appendix A). The latter observation may hint on the occurrence of certain miRNA editing events during processing. We also inspected the differential expression of protein-coding mRNA as well as long non-coding RNA in response to hypoxia in SW1736 by small RNA-seq analysis. Specifically, we found 12 protein-coding and 2 lncRNA transcripts to be up-regulated more than 2-fold *p* < 0.05 (Figure 3A; Appendix A). Surprisingly, no genes were significantly down-regulated within the given range (LFC < −1). However, hierarchical clustering strongly emphasized coordinated responses to hypoxia with at least 60 differentially expressed genes (−0.5 < LFC < 0.5) between the two groups (Figure 3B). Finally, gene ontology, KEGG and pathway analysis confirmed the enrichment of the deregulated transcripts in processes related to hypoxia response including the HIF-1 signaling pathway and glucose metabolism (Figure 3C,D).

### 2.4. Increase in miR-210-3p Release from SW1736 Cells in Hypoxia

Cells and conditioned cell culture media (CCM) were collected after 72 h of normoxic and hypoxic culture, and CCM was fractionated by size-exclusion chromatography (SEC) to obtain fractions enriched in extracellular vesicles (EVs), intermediate mixed EVs and proteins and extracellular proteins. Fractions were characterized per MISEV2018 guidelines [52]. The presence of EVs was verified by Western blot (WB) using EV-rich markers CD63, CD9, CD81, TSG101 and Syntenin, which were detected in EV-rich fractions but not in the later protein fractions (Figure 4B). The depletion of cellular markers such as GM130 and calnexin in extracellular fractions was confirmed relative to the cell source (Figure 4B). Single-particle interferometric reflectance imaging sensing (SP-IRIS) further confirmed EV-rich markers CD63, CD9 and CD81 relative to MIgG control (Figure 4C). Fractions were imaged by transmission electron microscopy (TEM), revealing that EVs ranged in diameter from ~50 to 300 nm in both normoxia and hypoxia in pooled EV-rich SEC fractions; EVs were not observed in protein-rich fractions (Figure 4D). We also assessed particle concentration and size by nano-flow cytometry (NFCM) (Figure 4E), finding no significant differences. qPCR for miR-210-3p showed that miR-210-3p was released from cells in all fractions during normoxia, and that this release increased, also across all extracellular fractions, by >2-fold in hypoxia (Figure 4F). Finally, immunoprecipitation confirmed the presence of miRNA carrier AGO2 in both mixed and protein fractions (Figure 4G).

## 3. Discussion

Highly aggressive tumors, which are often hypoxic, are frequently resistant to oxygen-dependent treatments such as chemotherapy and radiation. Thus, several direct and indirect methods to assess tumor hypoxia for targeted therapy, including miRNA expression, have been explored [53,54,55,56,57]. However, to our knowledge, miRNAs have not been extensively studied in hypoxic vs. non-hypoxic regions within ATC tumors. In this work, we examined miR-210, a direct HIF-1 target, as a potential marker of hypoxia in ATC [58,59]. Our results demonstrate that precursor miR-210 is down-regulated in ATC cell lines compared with PTC cells, but it is more robustly up-regulated in ATC in response to hypoxia. Additionally, we show that miR-210-3p can be detected extracellularly and is associated with EVs and extracellular AGO2.

miR-210 has paradoxically been described as both oncogenic and as a tumor suppressor in many studies [42,60,61,62]. It is believed to be involved in cell cycle regulation, metabolism and angiogenesis [63,64]. However, the exact role of miR-210-3p in tumorigenesis remains elusive. In this study, we also examined the differential expression of mRNA, but no down-regulated transcripts were identified. Polysome profiling by the proteomic analysis of ATC cell lines in hypoxia may be required to identify putative miR-210-3p targets in this cell type, as the use of a miR-210-3p mimic may result in off-target gene down-regulation.

Hypoxia can impact the differentiation state of cells by altering the expression of genes involved in various cellular pathways including cell cycle regulation and anaerobic metabolism [17]. Depending on the cell type, hypoxia can drive cells to a stem-like or differentiated state [65]. Furthermore, miR-210-3p may have a functional role in altering the differentiation state of cells in hypoxia [66]. It was interesting to observe lower basal levels of miR-210-3p in ATC cell lines SW1736 and C643, which are de-differentiated, compared with poorly and well-differentiated PTC cell lines BCPAP and TPC-1, respectively. This finding suggests that miR-210 might also be linked to tumor differentiation. It is believed that ATC can originate from PTC [67,68,69]; therefore, it is possible that miR-210 expression is impacted during de-differentiation. However, because miR-210 down-regulation was observed at both the precursor and mature level, it is unclear if the basal level expression differences are directly related to HIF-1 or miR-210 genetic or epigenetic alterations due to the differentiation levels of the thyroid cell lines. In either case, lower basal expression levels make miR-210 induction more pronounced and rapid in hypoxia; hence, miR-210-3p may be a better marker of hypoxia in ATC compared with PTC. Furthermore, miR-210-3p might also be a predictive marker of the de-differentiation of thyroid cancer cells.

We hypothesized that miR-210-3p might also be detected extracellularly in hypoxia, as miRNAs are remarkably stable in biological fluids such as blood plasma and are promising non-invasive biomarkers [29,32,34,35]. This is ascribed to their association with RNA carriers such as EVs and RNA-binding proteins, which offer protection from degradation [34,70]. In our study, we observed increased extracellular miR-210-3p in conditioned cell culture medium in hypoxia after at least 48 h, which further increased after 72 h. This suggests that, although cellular levels of miR-210 increase almost immediately in response to hypoxia, additional time is needed for the newly produced miRNAs to pass into the extracellular space and be detected there.

Apart from miR-210-3p, we documented the upregulation of various small RNAs aligning to more than 60 different protein-coding and lncRNA transcripts. Accordingly, gene ontology analysis revealed that multiple upregulated RNA fragments derived from protein-coding genes strongly related to hypoxia response including the HIF-1 signaling pathway and glucose metabolism. The mechanism of formation and the biological role of these mRNA-aligned small RNAs remains unknown; however, we suggest that they could be used as promising biomarkers for hypoxia in addition to miR-210-3p. The presence of those mRNA- and lncRNA-aligned short RNAs (mlnca-sRNA) could be explained by the uneven degradation of the parental transcripts that are retained in the small RNA fractions and the extracellular space.

In summary, our observations, albeit limited to in vitro studies, suggest that miR-210-3p might be a suitable miRNA marker of hypoxia in ATC.

## 4. Materials and Methods

### 4.1. Cell Culture

SW1736, C643, BCPAP and TPC-1 thyroid cell lines [71,72,73,74] were authenticated by the Johns Hopkins Genetic Resources Core Facility by Short Tandem Repeat (STR) profiling. Cells were cultured in RPMI-1640 medium (Gibco cat#11875-119, Grand Island, NY, USA) supplemented with: 10% Exosome-Depleted fetal bovine serum (Gibco A27208-01, Lot#2165597), 2mM L-glutamine (Thermo Fisher 25030-081, Waltham, MA, USA), 1% Non-Essential Amino Acids (Thermo Fisher cat#1140050), 10mM HEPES (Life Tech cat#5630-080, Newmarket, Australia), 100 U/mL Penicillin Streptomycin (Thermo Fisher cat#15140-122). Cells were cultured in a humidified 37 °C, 5% CO_2_ incubator under 21% O_2_ (normoxic) or in a 2% O_2_ (hypoxic) chamber, balanced with N_2_ using a compact O_2_/CO_2_ Controller ProOx c21 (Biospherix RS485, Parish, NY, USA) for 2, 4, 8, 24, 48, 72 or 96 h.

### 4.2. Cell Viability and Metabolic Activity Assays

Cell counts and viability were obtained using the Muse cell analyzer (EMD Millipore cat#1000-5175 serial#1846277, Burlington, MA, USA) and Muse Viability reagent (Millipore cat#MCH600103). Glucose concentrations from conditioned media were measured using a glucose monitoring system (bioreactor sciences GM-100 serial#15100005, Lawrenceville, GA, USA) and test strips (GMTS-50 bioreactor science).

### 4.3. Wound Healing Assay

Cells were seeded into 6-well plates and cultured until 100% confluent in normoxic conditions. Media was removed from wells and cells were washed twice gently with PBS. A 200 µL pipette was used to create a scratch. Cells were washed again, and a sharpie was used to mark a location for imaging. Images were taken at 0 h and 24 h during hypoxia.

### 4.4. EV Production and Separation

Cell culture conditioned medium was collected from cells and centrifuged at 1000× *g* for 5 min to pellet dead cells. The supernatant was centrifuged at 2800× *g* for 10 min to clear cell debris and apoptotic bodies. The 2800× *g* supernatant was then concentrated down to 500 µL at 3500× *g* using Centricon Plus-70 10kD cut-off concentrators (Millipore cat#UFC701008) for 1 h at 4 °C. A total of 500 µL of concentrated medium was fractionated over a qEV original 70 nm size-exclusion chromatography column according to the manufacturer’s instructions (IZON Science SP1). Pooled EV fractions (7–9) were concentrated down to ~100 µL at 4000× *g* using Amicon Ultra-2 Centrifugal Filter Units10kD cut-off concentrators (Millipore cat#UFC201024) at 4 °C.

### 4.5. Electron Microscopy

The EV and protein fractions were negatively stained and analyzed by transmission electron microscopy (TEM) at the Johns Hopkins Microscope Core Facility as previously described [75]. Samples were adsorbed (10 µL) to glow-discharged (EMS GloQube) carbon-coated 400 mesh copper grids (EMS), by flotation for 2 min. Grids were quickly blotted then rinsed in 3 drops (1 min each) of tris-buffered-saline (TBS). Grids were negatively stained in 2 consecutive drops of 1% uranyl acetate with tylose (1% UAT, double filtered, 0.22 µm filter), blotted then quickly aspirated to get a thin layer of stain covering the sample. Grids were imaged on a Phillips CM-120 TEM operating at 80 kV with an AMT XR80 CCD (8 megapixel).

### 4.6. Nano-Flow Cytometry Measurement (NFCM)

The NanoFCM Flow NanoAnalyzer was used to measure concentration and particle size following the manufacturer’s instructions and as described previously [75,76]. The instrument was calibrated separately for concentration and size using 250 nm Quality Control Nanospheres (NanoFCM) and a Silica Nanosphere Cocktail (NanoFCM cat#S16M-Exo) for detection of side scatter (SSC) of individual particles. Events were recorded for 1 min. Using the calibration curve, the flow rate and side scattering intensity were converted into corresponding particle concentrations and size.

### 4.7. Single Particle Interferometric Reflectance Imaging (SP-IRIS)

Measurements were performed as described previously [75]. A total of 50μL of EVs was diluted 1:1 in 1xIncubation Solution II and incubated at room temperature on ExoView Human Tetraspanin chips (Unchained Labs, Pleasanton, CA, Cat # 251-1000) printed with anti-human CD81 (JS-81), anti-human CD63 (H5C6), anti-human CD9 (HI9a) and anti-mouse IgG1 (MOPC-21). After incubation for 16 h, chips were washed with 1X Solution A I 4 times for 3 min each under gentle horizontal agitation at 460 rpm. Chips were then incubated for 1 h at room temperature with a fluorescent antibody cocktail of anti-human CD81 (JS-81, CF555), anti-human CD63 (H5C6, CF647) and anti-human CD9 (HI9a, CF488A) at a dilution of 1:1200 (v:v) in a 1:1 (v:v) mixture of 1X Solution A I and Blocking Solution II. The buffer was then exchanged to 1X Solution A I only, followed by 1 wash with 1X Solution A I, 3 washes with 1X Solution B I and 1 wash with water (3 min each at 460 rpm). Chips were immersed in water for approximately 10 s each and removed at a 45-degree angle to allow the liquid to vacate the chip. All reagents and antibodies were supplied by Unchained Labs (Pleasanton, CA, USA). Samples were diluted in 1X Incubation Solution II to load 50 μL of 4.0 × 10^8^ particles/mL, nominally, per chip. All chips were imaged in the ExoView R100 (Unchained Labs, Pleasanton, CA, USA) by interferometric reflectance imaging and fluorescent detection. Data were analyzed using ExoView Analyzer 3.1 Software (Unchained Labs). Fluorescent cutoffs were as follows: CD63 channel 200, CD81 channel 400, CD9 channel 400.

### 4.8. Cell and EV Lysis

Cells were removed from the incubator and immediately placed on ice. The medium was removed and immediately processed for EV purification. Cells were washed with ice-cold PBS and lysed in ice-cold 1X RIPA buffer (Cell Signaling 9806S, Danvers, MA, USA) with 1X Protease inhibitors (Santa Cruz sc-29131, Santa Cruz, CA, USA). Cells were incubated on ice for 10 min then transferred to tubes using cell scrapers. Lysates were cleared at 20,000× *g* for 10 min at 4 °C. The pellet was discarded. Total protein was measured using Pierce BCA Protein Assay Kit according to the manufacturer’s microplate protocol (Thermo Fisher 23225). A final concentration of 1X Laemmli sample buffer (Bio-Rad 161-0747) with 10% beta-mercaptoethanol (BME) (Bio-Rad 161-0710, Hercules, CA, USA) was added to 15 µg of total protein.

The EV and mixed samples were vortexed for 30s and incubated for 10 min at room temperature in 1X RIPA buffer (Cell Signaling 9806S) and 1X Protease inhibitors (Santa Cruz sc-29131) to lyse EVs. Total protein was measured using Pierce BCA Protein Assay Kit according to the manufacture’s microplate protocol (Thermo Fisher 23225). 1X Laemmli sample buffer (Bio-Rad 161-0747) was added to 1 µg of total protein.

### 4.9. Immunoprecipitation

Mixed and protein fractions were pre-cleared of IgG from FBS with 100 µL of protein G coated magnetic Dynabeads (Thermo Fisher 10003D Lot# 00715594) overnight at 4 °C with rotation. A total of 50µL of protein G beads was coated with 2ug of anti-AGO2 antibody (Sigma SAB4200085 Lot# 0000089486, St. Louis, MO, USA) according to the manufacturer’s instructions. The clearing protein G beads were removed from the sample and replaced with the anti-AGO2 coated beads overnight at 4 °C with rotation. The beads were washed three times with 1X PBST and resuspended in 1X Laemmli sample buffer 10% BME (Bio-Rad 161-0747).

### 4.10. Western Blot Analysis

Samples were heated to 95 °C for 5 min then separated alongside a spectra multi-color Ladder (Thermo Fisher 26634) through a 4–15% Tris-Glycine extended Stain-Free gel (Bio-Rad 5678085), at 100 V for 1.5 h using 1X Tris-Glycine SDS buffer (Bio-Rad 161-074). Gels were imaged with an EZ DocGel Stain-free imaging system (Bio-Rad 170-8274). Proteins were then transferred to a methanol activated (10s) PVDF membrane (Bio-Rad 1620177) at 100 V for 1 h in 1X Tris-Glycine buffer (Bio-Rad 161-0734) at 4 °C. The PVDF membrane was blocked for 1 h in blocking buffer (1XPBS (Gibco 14190-144), 0.05% Tween20 (Sigma-Aldrich 274348500), 5% blotting-Grade Blocker (Bio-Rad 170-6404)). The membrane was incubated with primary antibodies anti-CD63 (BD 556019 Lot#7341913) diluted 1:1000 and anti-CD81(sc-7637 Lot#C2318) diluted 1:500, anti-CD9 (BioLegend 312102 Lot#B351275, San Diego, CA, USA), ant-TSG101 (abcam ab228013 Lot#GR3306738-8, Cambridge, UK) diluted 1:500, anti-Syntenin (abcam ab133267 Lot# GR3375272-1) diluted 1:500, anti-GM130 (ab52649 Lot#GR3427322-2, anti-Calnexin (ab22595)) diluted 1:1000, anti-Albumin (abcam ab28405 Lot#GR3367930-2) diluted 1:1000, or anti Hif-1a (Cayman 10006421) diluted 1:200, anti-AGO2 (Sigma SAB4200085 Lot# 0000089486) diluted 1:1000 and anti-beta-actin (Sigma A1978) diluted 1:10,000 in blocking buffer overnight at 4 °C with rotation. The membrane was washed 3x for 5 min with rotation in blocking buffer. Secondary anti-mouse-HRP (Santacruz sc-516102 Lot#c1419) was diluted 1:10,000 or anti-rabbit-HRP (Dako P0448, Glostrup, Denmark) 1:1000 in blocking buffer and incubated for 1h at room temperature. Membranes were washed 3X in blocking buffer then 2X in 1XPBS 0.05% Tween20. Membranes were then incubated with Super Signal Chemiluminescent Substrate (Thermo 34580) for 5 min with gentle rotation and imaged by iBright FL1000 (Invitrogen, Waltham, MA, USA). For HIF-1a, band intensity in normoxia was normalized to hypoxia using Image J software.

### 4.11. RNA Isolation

Total RNA was extracted from cells using mirVana miRNA isolation kit (Ambion cat#AM1560, Austin, TX, USA) following the manufacturer’s protocol for adherent cells. Total RNA was extracted from size-exclusion chromatography (SEC) EV and protein fractions using miRNeasy serum/plasma kit (Qiagen 1071073 Lot#160020206, Hilden, Germany) after adding 1.0 × 10^6^ copies/µL of cel-miR-39 exogenous spike-in control (Qiagen 219610 Lot#157036035), according to the manufacturer’s instructions. The RNA concentration and purity were measured by NanoDrop (Thermo Fisher).

### 4.12. Small RNA Library Construction and Analysis

The ligation-independent Capture and Amplification by Tailing and Switching (CATS) small RNA-seq method was used to profile cellular small RNA originally described in [77]. Libraries were constructed with CATS RNA-seq Kit (Diagenode C05010041 Lot#4) following the manufacturer’s instructions. The remaining NGS library primers and other products shorter than 100 bp were removed by AMPureXP beads (Beckman, Brea, CA, USA). Quality control was assessed by Agilent bioanalyzer high sensitivity assay (Agilent, Santa Clara, CA, USA). Libraries were then additionally size-selected between 160 and 180 bp by BluePippin (Sage Science, Beverly, MA, USA). Prior to sequencing, libraries were spiked with 20% PhiX Control v3 (Illumina 1501766, San Diego, CA, USA) then run using the NovaSeq Illumina sequencing platform by the Johns Hopkins Microarray and Deep Sequencing Core. Sequencing quality control was performed using FASTQC (Babraham Bioinformatics, Cambridge, UK) followed by adapter and PolyA trimming with Cutadapt 1.17. The reads were first aligned to various housekeeping small non-coding RNA references including rRNA, tRNA, RN7S, snRNA, snoRNA, scaRNA, VT-RNA and Y-RNA (custom-curated from NCBI RefSeq and GENCODE). All reads that did not map to the aforementioned RNAs were sequentially aligned to mature miRNA (miRBase 22 release), pre-miRNA (miRBase 22 release), protein-coding mRNA transcripts and long non-coding RNAs (custom-curated references from GENCODE v28). The numbers of reads mapped to each RNA transcript type were extracted using eXpress software based on a previous publication [78]. All reads mapped to minor transcripts, pseudogenes and non-protein-coding parts of mRNAs were not included in the final analysis. The intronic and intergenic small RNA reads (combined) were extracted by mapping the remaining unaligned reads to hg38 genome reference. The differential expression analysis was performed by edgeR and limma packages as described in [54] using raw count tables as an input. Reads distribution over certain transcripts was visualized using Integrative Genomics Viewer (IGV). The GO-terms enrichment was assessed using the Enrichr knowledgebases as described previously [79] and visualized by the ggpot2 package.

### 4.13. RT-qPCR

Two step RT-qPCR was performed for miRNA analysis starting with 2 µL of RNA input from cells, EVs and Ex-protein fractions using TaqMan microRNA Reverse transcription kit (ABI 4366597 Lot#00636931), TaqMan miRNA stem-loop RT primers/qPCR primer probe set (ABI 4427975): (cel-miR-39 ID# 000200 Lot#P180110-003B10, miR-16 ID# 000391 Lot#P171018-000H05, miR-210-3p ID#000512) and TaqMan Universal Master Mix II, no UNG (ABI 4440040 Lot#1802074) as described in manufacturer’s protocol. For mRNA analysis, high-capacity cDNA synthesis kit (ABI cat#4368813), TaqMan qPCR primer probe set ABI: (HIF-1α ID#Hs00153153_m1, GAPDH ID# Hs02786624_g1, Beta-Actin ID# Hs01060665_g1) and TaqMan Universal Master Mix II, no UNG were used following kit protocol. qPCR was run on a CFX96 Real-time System (Bio-Rad). miR-210 was normalized to miR-16 and cel-miR-39 using the 2^−ΔΔCT^ method. HiF-1α was normalized to the average of GAPDH and Beta-actin Pooled by 2^−ΔΔCT^.

## Figures and Tables

**Figure 1 ijms-24-04507-f001:**
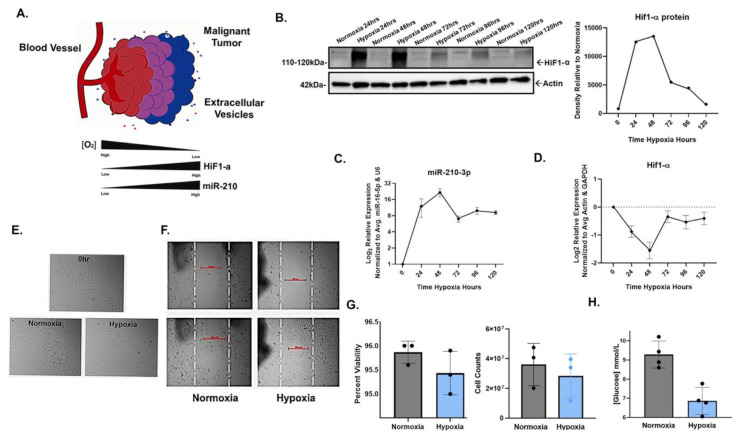
HIF1-α-induced miR-210-3p expression during hypoxic culture of SW1736 cells. (**A**). Cartoon illustration of hypoxic malignant tumor. In low oxygen conditions, miR-210 expression is regulated by HIF. (**B**). HIF1-α stabilization verified by Western blot (WB) and densitometry analysis after 24, 48, 72, 96 and 120 h of hypoxia (2% O_2_) or normoxia (21% O_2_). Actin was used as a loading control. (**C**). RT-qPCR analysis of mir-210-3p expression in hypoxia vs. normoxia at 24, 48, 72, 96 and 120 h of hypoxia relative to normoxia. Error bars represent the standard deviation of three biological repeats. (**D**). RT-qPCR analysis of HIF-α expression relative to normoxia (**E**). Images of SW1736 cells before and after 72 h of normoxia or hypoxia. (**F**). Wound healing assay of SW1736. (**G**). Percent viability of SW1736 cells after 72 h of normoxia or hypoxia. SW1736 cell counts after 72 h of normoxia or hypoxia. (**H**). SW1736 cell culture media glucose concentration after 72 h of normoxia or hypoxia.

**Figure 2 ijms-24-04507-f002:**
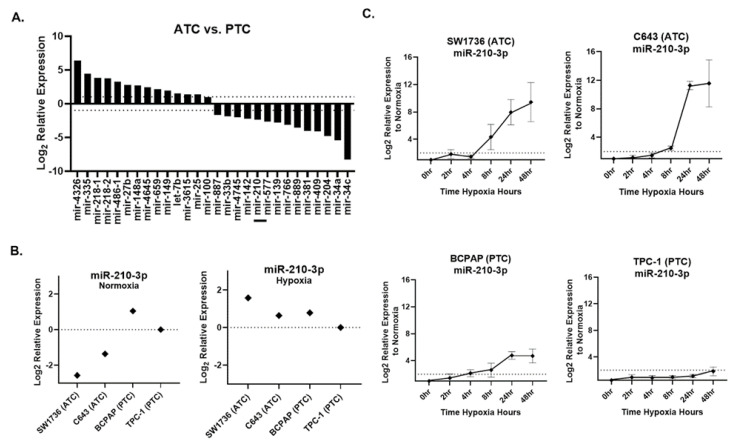
miR-210 expression in ATC and PTC cell lines. (**A**). The RNA-seq analysis of differentially expressed pre-miRNAs between anaplastic thyroid cancer (ATC) cell lines SW1736 and C643 and papillary thyroid cancer (PTC) cell lines BCPAP and TPC-1. Log_2_ fold change > ±1. *q* value < 0.05 n = 3. Bar included to highlight miR-210. (**B**). RT-qPCR of miR-210 basal expression and hypoxia (2%O_2_)-induced expression in ATC vs. PTC cell lines. (**C**). The ATC vs. PTC qPCR analysis of miR-210-3p expression during hypoxia time course study at 0, 2, 4, 8, 24, 48 h hypoxia.

**Figure 3 ijms-24-04507-f003:**
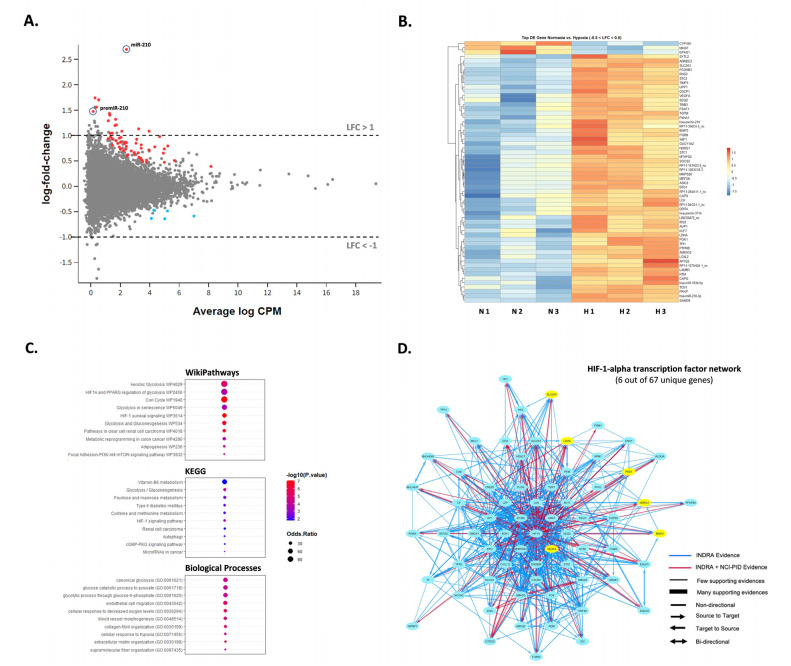
SW1736 cellular miRNA expression in hypoxia at 72 h. (**A**). The MD plot showing differential expression of miR-210-3p, pre-miR-210 as well as mRNA and lncRNA transcripts in hypoxia vs. normoxia datasets. Significantly deregulated genes (adj. *p*-value < 0.05) are shown in color. (**B**). Heatmap of the normalized read counts for all differentially expressed transcripts with −0.5 < LFC < 0.5 and adj. *p*-value < 0.05 (*n* = 66 in total). (**C**). Gene ontology, KEGG and pathway analysis of the deregulated genes in normal vs. hypoxic cells. (**D**). The NCI-Nature Pathway Interaction Database (PID) image visualizing HIF-1-alpha transcription network with the indicated protein-coding genes differentially expressed in datasets from hypoxic cells (in yellow).

**Figure 4 ijms-24-04507-f004:**
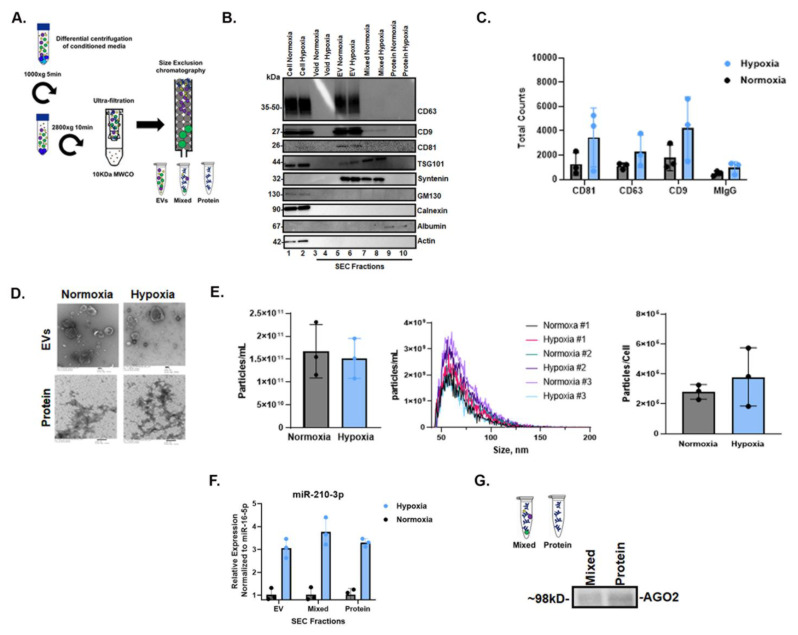
Characterization of cellular and extracellular fractions and AGO2 detection in hypoxic SW1736 culture. (**A**). Workflow diagram schematic of EV and protein separation from SW1736 cell culture media (CCM) by differential centrifugation and size-exclusion chromatography (SEC). Three independent experiments were performed. (**B**). Western blot analysis of tetraspanin CD63 and CD81 expression in SW1736 cell lysates and enrichment in hypoxic and normoxic SEC fractions after 72 h. (**C**). SP-IRIS detection of tetraspanins CD81, CD63 and CD9 in hypoxic and normoxic EV-enriched SEC fractions after 72 h. (**D**). Transmission electron micrograph (EM) of hypoxic and normoxic EV and protein SEC fractions after 72 h. (**E**). The SEC EV fraction particle counts per cell from normoxia and hypoxia. Data points represent three independent experiments by nano-flow cytometry. (**F**). miR-210-3p SEC pooled EV, mixed and protein qPCR analysis after 72 h hypoxia or normoxia, derived from 100 mLs of culture media. Data points represent three independent experiments. (**G**). AGO2 detection in SEC mixed and protein fractions by immunoprecipitation followed by Western blot.

## Data Availability

All relevant experimental details and data were submitted to the EV-TRACK knowledgebase (EV-TRACK ID: EV200090) [78] and Gene Expression Omnibus (GEO), accession numbers GSE207677 and GSE212703.

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
