# Peer review of "miR-210 Expression Is Strongly Hypoxia-Induced in Anaplastic Thyroid Cancer Cell Lines and Is Associated with Extracellular Vesicles and Argonaute-2"

_ijms, 2023, doi:10.3390/ijms24054507_

Round 1
Reviewer 1 Report
miR-210 expression is strongly hypoxia-induced in anaplastic thyroid cancer cell lines and is associated with extracellular vesicles & Argonaute-2.
This study evaluated miR-210-3p as a marker of hypoxia in papillary and anaplastic thyroid cancer cell lines.
Overall, this is a clear and concise manuscript. The study is relevant and documented, well conducted and argumented by the results of its well conducted laboratory experiments; it is well written and organized to elucidate the current knowledge and advances in the field as well as the past and ongoing works. However, I noted a few concerns which authors could address to improve the overall relevance of the findings:
The study of miR-210-3p as a marker of hypoxia is not new, as previous other studies have reported its up-regulated expression in various hypoxic tumor cell types and its association with a poor prognosis in various tumors. However, the novelty of the study resides in the evaluation of miR-210-3p as a marker of hypoxia in anaplastic and well differentiated thyroid cancer cell lines in hypoxic vs. non-hypoxic conditions.
The authors observed lower basal levels of miR-210-3p in de-differentiated ATC cell lines SW1736 and C643, compared with poorly differentiated PTC cell line BCPAP and well differentiated cell line TPC-1. It would be worth discussing more the results represented in figure 2C on how the higher expression level of miR-210-3p and the shorter exposure time to hypoxia was observed relative to the differentiation levels of the cell lines. This would make it even more interesting to compare the present results with the basal levels of miR-210-3p and the levels following hypoxic conditions in medullary thyroid carcinoma cells which are less differentiated than PTC cells but have also been reported to undergo fatal anaplastic de-differentiation, as these observations could provide further insights into the potential role of miR210-3p as predictive for the de-differentiation of thyroid cancer cells with hypoxia having a pivotal role in the process.
Author Response
Dear Reviewer,
Thank you for your comments on our study. We appreciate the time and effort you have taken to review our work.
- We followed the Reviewer’s suggestion to discuss Figure 2C results, which describe the higher expression level of miR-210-3p in hypoxia a shorter period of time in ATC cell lines relative to PTC cell lines. In the discussion section of the manuscript, we describe how hypoxia can impact the differentiation state of cells and how miR-210 may play a role and may be a potential marker for de-differentiation in thyroid cancer.
- Medullary cancer arises from Thyroid C-cells, a different lineage than PTC and FTC (follicular cells), which both arise from thyroid epithelial cells (thyrocytes). Most ATCs are derived from PTC & FTC. Since they are much more common than MTC, we focused on the follicular epithelial-derived cell lines. Investigating MTC lines may add valuable insights, but is beyond the scope of this paper.
Reviewer 2 Report
Some minor notes.
1. When describing the quantitative characteristics of extracellular vesicles, it is recommended to use the NTA method (as recommended by the ISEV standard) Although it is not mandatory when using the light scatter method, it is highly desirable.
2. It is a little strange that no significantly down-regulated mRNA was detected in the differential expression analysis, which may indicate possible problems with the differential expression analysis
3. The caption of Figure 3 (c) Heatmap of the normalized read counts 129 for all differentially expressed mRNAs with base Mean > 10, log2Fold Change > +/-1 and raw p value 130 < 0.05 (n=63 in total) indicates that raw p value was used to assess statistical significance. This is unacceptable, since the ajusted p value (FDR) should be used, as stated earlier in the article.
Author Response
Dear Reviewer,
Thank you for your comments on our study. We appreciate the time and effort you have taken to review our work.
Regarding the Reviewer’s recommendation to use NTA instead of the orthogonal, Nanoflow cytometry (NFCM), instrument to quantitate extracellular vesicles, we acknowledge that both techniques have advantages and limitations. NTA is well established and is recommended in the MISEV2018 guidelines. However, in our study we chose to use NFCM, which is a flow-based system to measure individual nanoparticles within the EV size range (40-1000nm). In previous studies by our lab group, we show that NFCM is more sensitive than NTA for small EVs (Arab et al J Extracell Vesicles 2021), indicating that NFCM can be a suitable alternative to NTA for this application. Additionally, NFCM requires less sample volume (50µL vs. 1mL) for particle quantification and size distribution, which enabled us to further characterize particles by electron microscopy & western blot.
To address the Reviewer’s concern regarding no significantly down-regulated mRNA transcripts in our Hypoxia vs. Normoxia differential expression analysis, we compared the EdgeR/limma DE software analysis tool to our DESeq2 analysis. EdgeR/limma confirmed the DESeq results and showed no down-regulated genes if using cut-off -1 < LFC 1. We have included the analysis for both approaches in the manuscript.
We appreciate the Reviewer’s concern and have made the necessary corrections using adjusted p value instead of raw p value to generate our heatmap in Figure 3 (c).